# Healthcare Providers’ Knowledge and Attitude Towards Abortions in Thailand: A Pre-Post Evaluation of Trainings on Safe Abortion

**DOI:** 10.3390/ijerph17093198

**Published:** 2020-05-04

**Authors:** Rugsapon Sanitya, Aniqa Islam Marshall, Nithiwat Saengruang, Sataporn Julchoo, Pigunkaew Sinam, Rapeepong Suphanchaimat, Mathudara Phaiyarom, Viroj Tangcharoensathien, Nongluk Boonthai, Kamheang Chaturachinda

**Affiliations:** 1International Health Policy Program, Ministry of Public Health, Nonthaburi 11000, Thailand; rugsapon@ihpp.thaigov.net (Ru.S.); nithiwat@ihpp.thaigov.net (N.S.); sataporn@ihpp.thaigov.net (S.J.); pigunkaew@ihpp.thaigov.net (P.S.); rapeepong@ihpp.thaigov.net (Ra.S.); mathudara@ihpp.thaigov.net (M.P.); viroj@ihpp.thaigov.net (V.T.); 2Division of Epidemiology, Department of Disease Control, Ministry of Public Health, Nonthaburi 11000, Thailand; 3Women’s Health and Reproductive Right and Foundation of Thailand, Bangkok 10210, Thailand; nonglukb@hotmail.com (N.B.); gumhang@hotmail.com (K.C.)

**Keywords:** abortion, training, health professionals, unplanned pregnancy, Thailand, pre-post evaluation

## Abstract

Although physicians in Thailand can carry out abortions legally, unsafe abortion rates remain high and have serious consequences for women’s health. Training programs for healthcare providers on the ‘Care of unplanned and adolescent pregnancies for the prevention of unsafe abortions’ have been implemented in Thailand with the aim of providing information and challenging negative attitudes about abortions. This study investigated the participants of the training courses in order to: (i) evaluate their knowledge and attitudes towards safe abortions; and (ii) investigate the factors that determine their knowledge and attitudes. A pre-post study design was applied. Descriptive statistics were calculated to provide an overview of the data. Bivariate analysis, a Wilcoxon signed rank test and a multivariable analysis using multiple linear regression were applied to determine the changes in attitudes and assess the likelihood of behaviour change towards adolescents and women experiencing unplanned pregnancy and abortions, according to demographic and professional characteristics. Having had the training, healthcare providers’ change in attitudes towards adolescents and women experiencing unplanned pregnancies and abortions were found to be 0.67 points for the nine responses of attitudes and 0.79 points for the 14 responses on various abortion scenarios. Changes in attitude were significantly different among the varying health professional types, with non-doctors increasing by 0.53 points, non-obstetricians and non-gynaecologists increasing by 0.46 points and obstetricians and gynaecologists (OBGYN) increasing by 0.32 points. Positive attitudes towards unplanned pregnancies and unsafe abortions and attitudes towards abortion scenarios significantly increased. The career type of the health professional was a significant factor in improving attitudes. The training program was more effective among non-doctor healthcare providers. Therefore, non-doctors could be the target population for training in the future.

## 1. Introduction

Annually worldwide, approximately 42 million women with unintended pregnancies choose to undergo abortion. Of these, approximately 20 million procedures are classified as unsafe abortions [1]. An unsafe abortion is defined as the “termination of an unwanted pregnancy either by persons lacking the necessary skills or in an environment that does not conform to minimal medical standards or both” [2]. This includes less-safe abortions, which are conducted using outdated methods by a trained provider or conducted using safe methods but not by a trained individual, as well as least safe abortions which are conducted using dangerous methods by untrained individuals [3]. Safe abortion encompasses clinical practice on abortion, such as the use of manual vacuum aspiration (MVA) and many wider social determinants including abortion-related laws and regulations. Women often resort to unsafe abortions due to barriers in accessing safe abortions, such as restrictive laws, unavailability of services, high cost, stigma, unnecessary requirements to obtain services and conscientious objections of healthcare providers [4]. Unsafe abortions can result in severe maternal consequences, including chronic health complications and disabilities, and are one of the leading causes of maternal death [1]. The majority of unsafe abortions, 97% globally, occur in developing countries in Africa, Latin America and Asia [3]. According to the World Health Organization, “As a preventable cause of maternal mortality and morbidity, unsafe abortions must be dealt with” [5].

Maternal health in Thailand has continued to suffer from unsafe abortion-related complications. A study conducted by the Thai Department of Health, found that 28.5% of hospital admissions from 787 government hospitals were a result of induced abortions. One third of these cases developed serious complications, for which over half had undergone abortions by unqualified healthcare providers [6]. Another study conducted in a Thai public hospital found that of all the women admitted for the treatment of complications from abortions, 36.8% underwent unsafe abortions [7]. Similarly, a study in the South of Thailand reported that unsafe abortions accounted for 35.7% of all abortions and were significantly associated with maternal, financial or family problems [8]. The fatality rate of abortions in Thailand is 300 per 100,000 abortions; this is a high rate compared with the fatality rate of 1 per 100,000 abortions in developed countries [9]. Adolescents are particularly vulnerable, as they are more likely to seek unsafe abortions and experience severe complications [10]. In Thailand, approximately 25.9% of all pregnancies are among adolescents, of which 14.4% result in abortions, making up 18% of the total abortions in the country [11]. The adolescent pregnancy rate in Thailand is 39 per 1000 women and according to the Division of Strategy and Planning of the Ministry of Public Health (MOPH) approximately half of adolescent pregnancies are at a high risk of undergoing unsafe abortions [12]. 

The Thai Penal Code 305 and the Thai Medical Council Regulation 2003 (BE 2548) permits abortions to be carried out by qualified physicians in certain conditions, including where there is risk to the health and life of the mother or child and pregnancies resulting from rape or incest. However, many women still face barriers to safe abortion services for various reasons, including lack of awareness of abortion services, societal stigma and taboo and the perception that abortion is absolutely illegal [9,13]. Healthcare provider attitudes towards abortions are also one of the major barriers preventing women from accessing safe abortion services in Thailand and therefore healthcare providers play a decisive role in ensuring safe abortions. [14]. Providers with negative attitudes towards abortion can lead women with unplanned pregnancies to risk their life and seek unsafe abortions conducted unlawfully by untrained health personnel [9,15]. Therefore, to rectify this problem, efforts to change the notion of abortion among healthcare providers is necessary. 

The Women’s Health and Reproductive Rights Foundation of Thailand (WHRRF), a non-profit organization that advocates the health and reproductive rights of women in Thailand, was established in 1998. The organization has been working in collaboration with the Thai Health Foundation, the Royal Thai College of Obstetricians and Gynaecologists (RTCOG) and the Thai Medical Council (TMC) to advocate improved health and standards of quality health services for women in the country, including access to safe abortion services. The WHRRF has been conducting a training program for healthcare professionals on the ‘Care of pregnant adolescents and women for the prevention of unsafe abortions’, to provide information and shift negative attitudes. Over the period of August 2017 to April 2018, three cohorts of health professionals participated in the program, held at the Royal Thai College of Obstetricians and Gynaecologists in Bangkok. The trainings were organized by the president and vice-president of the RTCOG as well as the president, vice-president and the secretary of the WHRRF and the advisor to the Thai Medical Council. The health professionals were invited to attend the training following a formal invitation letter sent to each Provincial Health Office by the WHRRF. Medical officers from the Department of Obstetricians and Gynaecologists of Ramathibodi Faculty of Medicines and Prince Songkla Faculty of Medicines were invited to conduct the training. The three-day training consisted of three modules. The first module explained the current global and national situation on abortion, including the debates on abortions, the effect of unsafe abortions on the socio-economic situation and health of women, the situations in which unsafe abortions occur, the risk of unsafe abortions and the benefits of access to safe abortions. The second module explained the methods used to provide safe abortions, with a focus on the use of manual vacuum aspiration (MVA) [16]. The final module demonstrated the method by instructors for the correct application of MVA’s and allowed participants to practice using MVAs on simulation models. Despite the active participation of the program, the impact of the program on the attendees was never evaluated. This study therefore aimed to: (i) evaluate the knowledge and attitudes towards safe abortion among the training courses participants and (ii) investigate the factors that determined the knowledge and attitudes among the training course participants. 

## 2. Methods

### 2.1. Study Design and Participants

A pre-post study design was applied. A self-administered survey was filled out by each participant before and after the training. The time used for responding to the questionnaire was about 10–15 min for pre-test and for post-test. The course attendees were asked to return the filled questionnaire form to the course facilitator. Data from health professionals who attended the training program on the ‘Care of pregnant adolescents and women for the prevention of unsafe abortions’ between August 2017 and April 2018 were included. 

### 2.2. Ethics Approval

This study is part of the routine monitoring system by IHPP on progress and access to sexual and reproductive health. IHPP is a research institute of the Ministry of Public Health, and therefore it was not necessary to obtain ethics approval. Despite this, the researchers strictly followed ethical standards where all individual information was strictly kept confidential and not reported in the paper.

### 2.3. Data Collection and Measures

Data from surveys conducted by the Women’s Health and Reproductive Rights Foundation of Thailand were obtained. The survey was composed of four main parts: (1) demographic characteristics; (2) work experience with adolescents and women who have had unplanned pregnancies and undergone abortions; (3) perceptions towards adolescents and women with unplanned pregnancies and unsafe abortions (9 sub-questions); and (4) scenarios on abortions (14 sub-questions). Details of all the sub-sections are presented in the results section.

The demographic measures comprised sex (male, female), age (years), type of health profession (doctor, nurse, pharmacist and welfare workers) and the specialization of doctors (general practice, obstetrics and gynaecology, family medicine, preventative medicine, among others). The measures of experience of working with adolescents and women who had experienced unplanned pregnancies and undergone abortions included: (a) counselling experience with adolescents and women who had unplanned pregnancies (ever, never); (b) treating adolescents and women who had undergone abortions already (ever, never); (c) regulations and guidelines on the termination of pregnancies (know, do not know); (d) knowledge of manual vacuum aspiration (MVA) as a tool for safe abortion (know, do not know; ever seen, ever heard, never seen, never heard); and (f) experience in using manual vacuum aspiration (MVA) (yes, no). The measures of the attitudes of health professionals were rated on a five-point Likert scale ranging from 1 representing ‘strongly disagree’ to 5 ‘strongly agree’. Nine items measured the general attitudes towards adolescent pregnancy and fourteen items measured the attitudes towards various scenarios for when abortions may be performed.

### 2.4. Analysis

Pre- and post-surveys were matched using a participant code number that was provided to each participant when they registered to the training and recorded onto both the pre-survey and post-survey. Participants who did not complete both the pre- and post-surveys or whose information about demographic characteristics was incomplete were excluded from the analysis. All analyses were conducted using the Stata/IC version 14 (StataCorp LLC, Texas, TX, USA) and the statistical significance was assessed at alpha of 0.05. and the statistical significance was assessed at alpha of 0.05. 

### 2.5. Descriptive Statistics

Descriptive statistics were calculated for all the demographic characteristics and experiences of working with adolescents and women with unplanned pregnancies and having undergone abortions, including percentages for categorical variables, and mean and standard deviation (SD) for continuous variables. The mean, SD, median and interquartile range (IQR) were calculated for each item for participant attitudes towards adolescents and women with unplanned pregnancies and abortions and attitudes towards various abortion scenarios. 

### 2.6. Bivariate Analysis

A bivariate analysis was conducted to determine the significance of the change in attitudes towards adolescents and women with unplanned pregnancies and having undergone abortions as well as the attitudes towards various abortion scenarios. The Shapiro–Wilk test was used to examine distribution normality. The Wilcoxon signed rank test was applied to avoid assumptions of normal distribution needed for the paired sample t-tests. The Kruskal–Wallis rank test was applied to analyse the significance of the change in attitudes according to demographic characteristics and professional experiences, including sex, age, profession as well as knowledge and experience on abortion including aspects of regulations, treating, counselling and MVA. 

### 2.7. Multivariable Analysis

Multivariable analysis was applied to assess the likelihood of the change in attitudes towards adolescents and women with unplanned pregnancies and having undergone abortions as well as attitudes towards various abortion scenarios, using demographic and experience characteristics as predicator variables. Multiple linear regression was applied. The outcome variables were the change in the average score between the pre-test and post-test results on attitudes towards adolescents and women with unplanned pregnancies and having undergone unsafe abortions (Y1) and the response to abortion scenarios (Y2) (part 3 and part 4 of the questionnaire). The predictor variables were selected from the results of the bivariate analysis. Those with statistical significance at *p* < 0.2 were included in the multivariable analysis. The authors also conducted a kernel-based regularized least squares method, a special case of linear regression which allows the researchers to account for data with non-normal distribution, in order to assess whether the results would differ from conventional regression analysis. 

## 3. Results

A total of 325 participants attended the training programme: 99, 147 and 79 in the first, second and third batch, respectively. Of these, 250 completed both the pre- and post-test surveys and had a matching survey pair. The demographics and professional experiences section were incomplete for three participants, who were excluded (Figure 1). The data from 247 participants were included and analysed: 43.3% from the second training session, 30.8% from the first and the remaining 25.9% from the third. 

### 3.1. Demographics Characteristics and Work Experience

The participant age ranged from 23 to 58 years with an average of 35.9 years. The majority of participants were female (84.6) and were nurses (55.5%). Of the participants that were doctors (43.72%), 52.8% were general practitioners and 36.1% were obstetricians and gynaecologists. The survey found that 63.2% of all participants had prior experience of counselling for unplanned pregnancies and 80.6% had previously treated patients that had undergone abortions. 52.5% of participants had previous knowledge on the regulations of the Medical Council on Practices Regarding Termination of Medical Pregnancy. With regards to manual vacuum aspirations, 65.23% had knowledge of MVA, 64% had knowledge on the requirements of the use of MVA, 67.21% had previously seen an MVA and 50.21% had used an MVA. Details on the demographic characteristics and experiences of all the participants are summarized in Table 1.

### 3.2. Comparison of Pre-post Results on Attitudes towards Adolescents and Women Experiencing Unplanned Pregnancies and Unsafe Abortions

The median pre-test and post-test responses were found to be significantly different for each of the nine responses on attitudes towards unplanned pregnancies and unsafe abortions at *p* < 0.001 (Table 2). The median of the combined average of all nine responses was found to have significantly increased by 0.67 points at *p* < 0.001 (Figure 2).

The difference in the average responses for attitudes towards adolescents and women experiencing unplanned pregnancies and having undergone unsafe abortions was found to be significantly different according to career at *p* = 0.013. Although the median pre-test scores were equal for all career types, with an average score of 4, the median post-test scores highly varied (Appendix A). The greatest increase in positive change was found in non-doctor participants with an average increase of 0.53 points, compared with doctors who were not specialized in obstetrics and gynaecology with an average increase of 0.46 points, and obstetricians and gynaecologists with an average increase of 0.32 points. The differences in the average response were also found to be significant among participants who had prior knowledge of the regulations of the Medical Council on Practices Regarding the Termination of Medical Pregnancy (*p* = 0.0039). Those with prior knowledge had an average increase of 0.55 points, while those without prior knowledge increased by an average of 0.42 points (Table 3).

The predictor variables incorporated into the regression model included healthcare professions, prior knowledge of the regulations of the Medical Council on Practices Regarding Termination of Medical Pregnancy and experience of treating and counselling teenagers and women with unplanned pregnancies. The analysis determined that non-medical doctor health professionals were most likely to benefit from the training. Although both the multiple linear regression (Table 4) and the kernel-based regularized least squares method (Appendix A) showed similar trends, the results were only significant when using multiple linear regression (*p* = 0.041).

### 3.3. Comparison of Pre-post Results on Attitudes towards Various Scenarios for Abortions

The median pre-test and post-test responses were found to be significantly different for each of the 14 responses concerning attitudes towards abortion scenarios (Table 5) at *p* < 0.001. The median of the combined average of all 14 responses was found to have significantly increased by 0.79 points at *p* < 0.001 (Figure 2). The difference in the average responses to examples of scenarios on abortions was also found to be significantly different according to career at *p* ≤ 0.001. The most significant increase was found in non-doctor participants, with an average increase of 0.84 points, compared with doctors not specialized in obstetrics and gynaecology with an average increase of 0.59 points and then obstetricians and gynaecologists with an average increase of 0.54 points (Table 3). However, it is important to note that the pre-test scores of non-doctors were lower with a median score of 3.57, compared to the median scores of obstetricians and gynaecologists at 3.79 and other doctors with a score of 3.64 (Appendix A).

The predictor variables incorporated into the regression model included healthcare profession, prior knowledge of the regulations of the Medical Council on Practices Regarding Termination of Medical Pregnancy, and experience of counselling teenagers and women with unplanned pregnancies. Similarly to the analysis of attitudes towards adolescents and women experiencing unplanned pregnancies and having undergone unsafe abortions, the regression analysis determined that non-medical doctor health professionals were the most likely to benefit from the training. The results were found to be significant using both multiple linear regression, at *p* = 0.003 (Table 6) and kernel-based regularized least squares method, at *p* = 0.006 (Appendix A and Appendix A).

## 4. Discussion

The evidence indicated that following the implementation of the training programme, healthcare providers’ positive attitudes towards unplanned pregnancies and unsafe abortions significantly increased. Overall, career types of healthcare providers significantly contributed to changes in attitudes towards unplanned pregnancies and unsafe abortions. Detailed analysis suggested that non-obstetric and non-gynaecology doctors and supporting staff (such as nurses) tended to benefit most from the training. This was evident in the significant increase in their assessment scores relative to the scores for obstetrics and gynaecology doctors (as shown in Table 5 and Table 6).

Our findings were similar to findings from other studies. For instance, in Zimbabwe, a study found that supporting staff apart from doctors (such as a nurse, midwife, senior nurse or hospital administrator) play an important role in supporting women in accessing safe abortions [17]. A study by Cooper et al. gave a positive view on nurses’ and midwives’ attitudes towards abortion [18]. However, a systematic literature review on the perceptions and attitudes of healthcare providers in sub-Saharan Africa and Southeast Asia by Ulrilka et al. suggested that nurses and midwives disliked being involved in abortion services, and commonly reported hesitancy in providing these services. The nurses’ resistance to providing abortion services was a powerful barrier against access to safe abortion services, with nurses’ and midwives’ strong opposition to abortion affecting rural women in particular [19]. Such findings however contradicted the findings in this study, as the results did not present strong negative attitudes towards abortion among nurses and supporting staff. In contrast, the findings showed that nurses and supporting staff could be potential target groups for further trainings, as their scores were enhanced the most compared with other health professionals.

This study has some policy implications. Firstly, this kind of training on safe abortion is useful to wider health professionals, institutions and organizations, which in turn can play an important role in creating awareness of unsafe abortion and providing safe abortion services. Secondly, abortion is not only a matter for obstetrics and gynaecologists. The study found that non-obstetric doctors and support staff can play an important role. Finally, this study found that knowledge of regulations on abortions is quite low as less than 50% of the participants had adequate prior knowledge on these (Table 1). Therefore, the Royal Thai College of Obstetricians and Gynaecologists and the medical council should work together to communicate these regulations to health professionals and the wider public. 

Some limitations remain. Firstly, the participation in this training was voluntary and this therefore created a risk of selection bias. Those who opt out from this kind of training may not have similar favourable attitudes towards abortion. Secondly, the exclusions of participants undermined the statistical power as some observations were dropped (Figure 1). This may be a reason why some factors did not show statistical significance. However, researchers checked the demographic characteristics of the participants that were excluded and found no significant differences compared to the study sample. Thirdly, analyses by different methods can yield different results [20]. The questionnaire in section three and four applied a Likert scale to analyse the attitudes of participants. In this study we opted to use regression analysis; however, if researchers devised and used other analytical methods such as the chi squared or logistic analyses, they may yield different results. However, researchers found the results to be quite valid as the kernel-based regularized least squares method was applied and there was no marked difference in the results. Fourth, participants’ locations of work, either urban or rural settings, were not collected, and as the trainings were conducted only in Bangkok, it is expected that most participants were from the Bangkok metropolitan area. Therefore, differences in attitudes between urban and rural areas could not be clearly identified. Further studies should be conducted to analyse the urban and rural area differences in the attitudes by the location of their workplaces. Finally, this assessment is subjective. The results showing the positive attitudes of the participants who attended the training programme do not guarantee good practice in a real-life situation and the study sample does not necessarily reflect the viewpoints of all the health professionals in the country. Further research should assess real-life practices and attitudes, including both qualitative and quantitative components.

## 5. Conclusions

Following the training, the score for positive attitudes towards unplanned pregnancies and unsafe abortions as well as attitudes towards abortion scenarios significantly increased. The main determinant, which significantly contributed to positive attitudes towards unplanned pregnancies and unsafe abortions, was the career type of the healthcare provider. In particular, non-doctor health professionals were likely to benefit the most from this kind of training and could be the target population for training in the future. Further research using both qualitative and quantitative methods should be conducted to assess the attitudes and real-life practices of healthcare professionals concerning abortions in Thailand.

## Figures and Tables

**Figure 1 ijerph-17-03198-f001:**
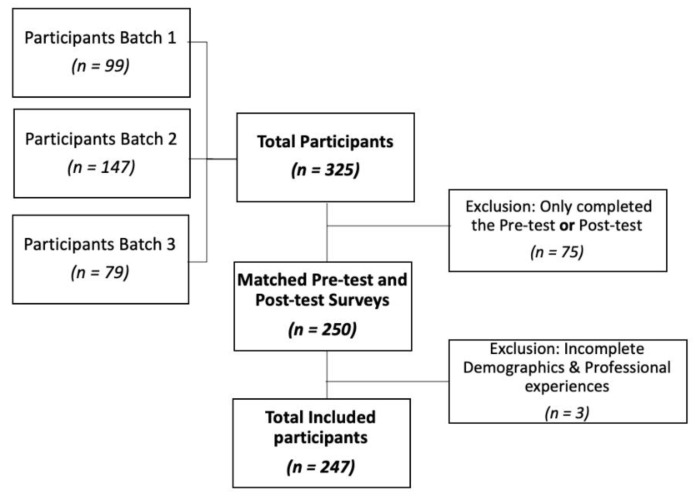
Flow diagram of the included participants.

**Figure 2 ijerph-17-03198-f002:**
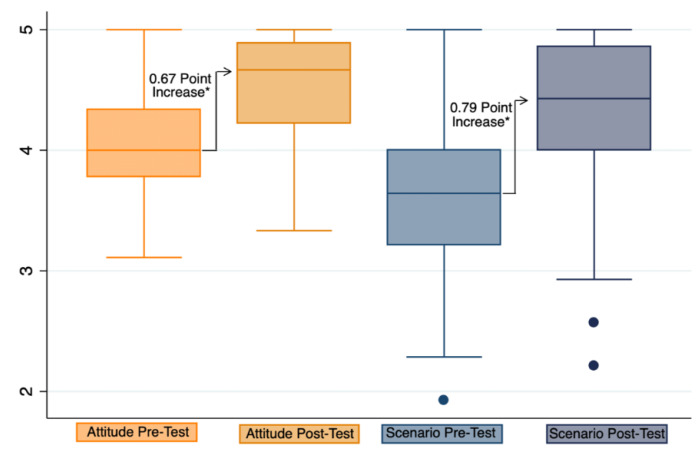
Box plot of the average pre-test and post-test responses. Note: Attitudes towards adolescents and women unplanned for pregnancy, and unsafe abortions. Scenario: Response to example of scenarios on abortions. * *p*-Value ≤ 0.001

**Table 1 ijerph-17-03198-t001:** Participant Characteristics.

Characteristics	Total (*n* = 247)
Sex (male/female) (%)	15.4/84.6
Age (Year) (Mean ± SD) [Min–Max]	35.9 ± 8.8 [23–58]
Profession (%)	
Doctor (*n* = 137) (% of total participants)	43.7
General (% of doctors)	52.8
Obstetrics and gynaecology (% of doctors)	36.1
Family medicine (% of doctors)	4.6
Preventative medicine (% of doctors)	1.9
Others (% of doctors)	4.6
Nurse (% of total participants)	55.5
Pharmacist (% of total participants)	0.4
Welfare workers (% of total participants)	0.4
Prior experience counselling for unplanned pregnancies (% with experience)	63.16
Prior experience treating for unplanned pregnancies (% with experience)	80.57
Knowledge of regulations of the Medical Council on the Practices Regarding the Termination of Medical Pregnancies (% with knowledge)	52.46
Manual vacuum aspirations (MVA)	
Know of MVA (%)	65.23
Knowledge of requirements for medical professionals for the use of MVA (%)	64.00
Seen MVA (%)	67.21
Used MVA (%)	50.21

**Table 2 ijerph-17-03198-t002:** The comparison of the pre- and the post-test attitudes towards adolescents and women experiencing unplanned pregnancies and unsafe abortions (*n* = 247).

Questions	Pre-Test	Post-Test
	Mean (SD), Median [IQR]	Mean (SD), Median [IQR]
1. At the present, unplanned pregnancies and unsafe abortions are a major public health problem that should be addressed.	4.62 (0.50),	4.76 (0.43),
	5 [1]	5 [0]
2. In your area, unplanned pregnancies and unsafe abortions are a major public health problem, that should be addressed.	4.10 (0.71),	4.39 (0.68),
	4 [1]	4 [1]
3. One reason for unsafe abortions is the limited options for pregnant women and the societal pressures pregnant women face.	4.02 (0.78),	4.64 (0.54),
	4 [1]	5 [1]
4. Family and society should help unplanned pregnancies.	4.50 (0.54),	4.74 (0.46),
	5 [1]	5 [1]
5. Women with unplanned pregnancies should have the right to decide and choose whether to continue or terminate the pregnancy.	4.00 (0.82),	4.61 (0.59),
	4 [1]	5 [1]
6. Women that need to terminate their pregnancies, according to the criteria set by the Regulations of Medical Council of Thailand, should receive safe abortion, with the same level of services and benefits as other health problems.	4.38 (0.59),	4.72 (0.46),
	4 [1]	5 [1]
7. Doctors and healthcare providers play a major role in addressing unplanned pregnancies and unsafe abortions.	4.49 (0.54),	4.72 (0.48),
	5 [1]	5 [1]
8. You are happy to help, advise and provide consultations regarding abortions and places to obtain safe abortion services for those with unplanned pregnancies.	4.05 (0.84),	4.53 (0.58),
	4 [1]	5 [1]
9. Thailand should allow the sale of medical abortion drugs as well as emergency contraceptives at pharmacies.	2.38 (1.24),	3.74 (1.20),
	2 [2]	4 [2]
Average Score of all questions	4.06 (0.42),	4.54(0.41),
	4 [1.6]	4.67 [0.6]

Note: All *p*-Values were ≤ 0.001 using the Wilcoxon signed rank test.

**Table 3 ijerph-17-03198-t003:** Differences in the pre- and the post-test changes in the responses according to demographic and professional characteristics.

Characteristics	Attitudes towards Adolescents and Women Experiencing Unplanned Pregnancies, and Unsafe Abortions	Change in Response to Examples of Scenarios on Abortions
Sex	*p* = 0.863	*p* = 0.217
Female	0.48 (0.43), 0.44 [0.55]	0.75 (0.58), 0.71 [0.71]
Male	0.46 (0.38), 0.44 [0.55]	0.61 (0.40), 0.64 [0.5]
Age	*p* = 0.821	*p* = 0.900
Less than 35	0.46 (0.44), 0.44 [0.55]	0.71 (0.52), 0.71 [0.64]
More than 35	0.49 (0.40), 0.44 [0.55]	0.74 (0.60), 0.71 [0.64]
Career	*p* = 0.013 *	*p* < 0.001 *
OBGYN	0.32 (0.38), 0.33 [0.44]	0.54 (0.46), 0.42 [0.64]
Other doctor	0.46 (0.38), 0.44 [0.44]	0.59 (0.44), 0.57 [0.57]
Non-doctor	0.53 (0.44), 0.55 [0.66]	0.84 (0.61), 0.86 [0.71]
Knowledge of MVA	*p* = 0.248	*p* = 0.507
Knowledge of regulation	*p* = 0.004 *	*p* = 0.189
Prior knowledge	0.55 (0.43), 0.55 [0.61]	
No prior knowledge	0.42 (0.39), 0.33 [0.55]	
Experience in treating	*p* = 0.071	*p* = 1.000
Experience in counselling	*p* = 0.087	*p* = 0.157
Experience in using MVA	*p* = 0.846	*p* = 0.698

Note: Mean (SD), Median [IQR]; * Statistically significant (Defined as *p* < 0.05); OBGYN: Obstetricians and gynaecologists.

**Table 4 ijerph-17-03198-t004:** Least-squares regression analysis on the attitudes towards adolescents and women experiencing unplanned pregnancies and having undergone unsafe abortions.

(Reference: OBGYN)	Coef. (95% CI)	*p*-Value
Non-OBGYN doctor	0.095 (−0.074, 0.264)	0.269
Non-doctor	0.165 (0.006, 0.323)	0.041
Prior knowledge of regulation	−0.067 (−0.182, 0.047)	0.247
Experience in treating	−0.102 (−0.236, 0.032)	0.134
Experience in counselling	0.002 (−0.002, 0.007)	0.317
Constant	0.475 (0.279, 0.671)	0.000

**Table 5 ijerph-17-03198-t005:** The comparison of the pre- and the post-test responses to the examples of abortion scenarios (*n* = 247).

Questions	Pre-Test	Post-Test
	Mean (SD), Median [IQR]	Mean (SD), Median [IQR]
1. If the pregnant woman has underlying diseases and the pregnancy poses serious harm to their health or life.	4.56 (0.59), 5 [1]	4.85 (0.39), 5 [0]
2. If the pregnant woman has physical or intellectual disabilities hindering their ability to care for themselves.	4.45 (0.66), 5 [1]	4.81 (0.49), 5 [0]
3. If the pregnant woman has HIV/AID.	3.40 (1.23), 3 [3]	3.74 (1.25), 4 [2]
4. If the pregnant woman has rubella.	4.03 (0.89), 4 [1]	4.50 (0.71), 5 [1]
5. If the foetus has anomalies that can result in being physically or intellectually disabled.	4.31 (0.81), 4 [1]	4.74 (0.52), 5 [0]
6. If the foetus has genetic disorders or serious diseases.	4.38 (0.81), 4 [1]	4.78 (0.49), 5 [0]
7. If the pregnant woman’s mental health is at risk.	3.71 (0.97), 4 [2]	4.53 (0.65), 5 [1]
8. If the pregnant woman is under the age of 15.	3.33 (1.07), 3 [1]	4.32 (0.83), 5 [1]
9. If the pregnant woman is under the age of 20 and still in school.	3.01 (1.04), 3 [2]	3.97 (0.93), 4 [2]
10. If the pregnancy is a result of rape.	4.34 (0.76), 4 [1]	4.78 (0.47), 5 [0]
11. If the pregnancy is a result of incest.	3.39 (1.03), 3 [1]	4.24 (0.91), 5 [1]
12. If the pregnancy is a result of contraceptive failure.	2.97 (1.15), 3 [2]	4.19 (0.87), 4 [1]
13. If the pregnant woman is facing economic problems.	2.76 (1.08), 3 [1]	4.11 (0.93), 4 [1]
14. If the pregnant woman is unmarried.	2.67 (1.11), 3 [1]	3.91 (0.99), 4 [2]
Average Score of all questions	3.67 (0.64), 3.64 [0.79]	4.39 (0.53), 4.43 [0.86]

Note: All *p*-values were ≤ 0.001 using the Wilcoxon signed rank test.

**Table 6 ijerph-17-03198-t006:** Least-squares regression analysis on the abortion scenarios.

(Reference: OBGYN)	Coef. (95% CI)	*p*-Value
Non-OBGYN doctor	0.064 (−0.159, 0.288)	0.573
Non-doctor	0.323 (0.112, 0.534)	0.003 *
Prior knowledge of regulation	0.041 (−0.109, 0.191)	0.54
Experience in counselling	0.003 (−0.004, 0.009)	0.437
Constant	0.489 (0.227, 0.751)	0.000

* Statistically significant (Defined as *p* < 0.05)

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
