# Peer review of "Healthcare Providers’ Knowledge and Attitude Towards Abortions in Thailand: A Pre-Post Evaluation of Trainings on Safe Abortion"

_ijerph, 2020, doi:10.3390/ijerph17093198_

Round 1
Reviewer 1 Report
It is a study of great interest and the topic is very necessary.
Some important information is missing, especially those related to the data collection procedure. I also recommend to better describe the training program.
It is also necessary a better description of the ethical considerations. ¿Did the participants know that their information would be anonymized? ¿Did they accept to participate in the study?
The results show differences between doctors and nurses. Has it also been studied gender differences?
It also would be interesting to include in the discussion the area in which the study was developed as well as the differences that other studies documented between rural and urban areas.
Author Response
- There are now considered to be three categories of abortion; safe, less safe, and least safe. The introduction should be updated to reflect those categories.
The definition of less safe and least safe abortions have been added in addition to the previously detailed definitions of safe and unsafe abortions; please see line 49-52 and a new reference 3 was inserted to the manuscript.
- The data on the level of unsafe abortion in Thailand is from a paper from 2004. Are there any more recent data available?
More recent data from Thailand were added, please see line 66-70, and two new references 7 and 8 were added.
- It is not clear to me how participants were recruited. Who was involved in the training?
Details on participant recruitment and training have been added, please see line 99-104.
- How were pre- and post-test surveys linked?
We used a participant identification number to match each individual; these ID number was given to each participant at the registration to training workshop, please see line 136-138.
- Changes in attitudes by various factors are not really univariate analyses, but, rather bivariate analyses.
Agree to change to bivariate analysis. please see line 149-150, and throughout the paper for consistency.
- It would helpful to have more description of the questions that make up the “attitudes towards adolescents and women with unplanned pregnancies and unsafe abortions”, as well as how “prior knowledge of the law” was determined.
The attributes related to “attitude towards adolescents and women with unplanned pregnancies and unsafe abortions” include 14 different scenarios as detailed in Table 5.
Participants rated each scenario on a five-point Likert scale of 1 to 5, with 5 being strongly agree that abortion should be provided in that situation. All questions in the survey were self-reported. On prior knowledge of the law, each participant answered yes/ no to whether they knew the regulations of the Medical Council on practices regarding the termination of medical pregnancies, prior to attending the training. We have presented these in Table 1, and also please see Line 119-120.
- I would be interested to see the mean scores by profession pre- and post. I am curious if the non-doctors started lower and then their score increased to a level that was closer to the doctor’s scores, or if they were closer. Were the ObGyns and other doctors less impacted because they already had a higher level of positive attitudes?
Many thanks for the question, we clarified that the pre-test median scores of non-doctors were slightly lower, 3.57 compared to the OBGYN, 3.79 and other doctors, 3.64.
Information on pre-test and post-test scores by profession has been added. We have added figures that show mean scores by professionals in supplementary file 1, additionally, please see Line 206-207 and 236-239.
- It is stated in the discussions section that prior knowledge on the law is significantly associated with improving attitudes towards abortion. Where is this indicated in the results sections? In the two regression tables, I see that prior knowledge is not associated.
Prior knowledge on the law showed significant association using bivariate analysis. Those with prior knowledge had an average increase of 0.55 points, while those without prior knowledge increased by an average of 0.42 points. This was presented in line 211-215 and table 3, yet no significance shown in the multivariable analysis. We have reformulated the discussion to reflect this evidence. Please see Line 255
Our intention regarding the discussion in Line 278 - 280 on this issue is to point out further policy implication to improve knowledge related to abortion laws and regulation among participants; as evidence from this study showsthat prior knowledge is the most important determinant for attitude improvement toward abortion services. This is because the level of knowledge of regulations on abortions is quite low (as less than 50% of participants had adequate prior knowledge on abortion laws).
- It is written, “Such findings however contradict the findings in this study, as the results did not present strong negative attitudes towards abortion among nurses and supporting staff. In contrast, findings showed that nurses and supporting staff could be potential target groups for further trainings, as their scores were enhanced the most compared with other health professionals.” I do not quite agree with this assessment of the findings. Firstly, without knowing how the participants were selected for the training (did they self-select, in which case, they might be already more open to abortion), and without seeing the levels pre-training, it is impossible to state whether the findings presented here support or contradict these previous findings.
Additional information on participant selection has been added. Please see line 98 -104. It should be noted that there were no differences in the selection of doctors compared to non-doctors into the training program, therefore, this limits the selection bias of the training participants. Additionally, this study compared the change in scores, which takes into account pre-test scores, and in our regression model, we have adjusted for prior knowledge on the existing laws and regulations as well as participant experience in treating and counselling on unplanned pregnancies and abortions.
Therefore, findings from this study was grounded by its evidence that non-doctors are a potential target group for these trainings as they showed the most significant improvement. This finding contradicts the study which found that nurses’ and midwives’ were strongly opposed and resistant; for which there are great differences in socio-economic, political, cultural, beliefs and religious and training of health personnel across countries.
Reviewer 2 Report
There are now considered to be three categories of abortion; safe, less safe, and least safe. The introduction should be updated to reflect those categories.
The data on the level of unsafe abortion in Thailand is from a paper from 2004. Are there any more recent data available?
It is not clear to me how participants were recruited. Who was involved in the training?
How were pre- and post-test surveys linked?
Changes in attitudes by various factors are not really univariate analyses, but, rather bivariate analyses.
It would helpful to have more description of the questions that make up the “attitudes towards adolescents and women with unplanned pregnancies and unsafe abortions”, as well as how “prior knowledge of the law” was determined.
I would be interested to see the mean scores by profession pre- and post. I am curious if the non-doctors started lower and then their score increased to a level that was closer to the doctor’s scores, or if they were closer. Were the ObGyns and other doctors less impacted because they already had a higher level of positive attitudes?
It is stated in the discussions section that prior knowledge on the law is significantly associated with improving attitudes towards abortion. Where is this indicated in the results sections? In the two regression tables, I see that prior knowledge is not associated.
It is written, “Such findings however contradict the findings in this study, as the results did not present strong negative attitudes towards abortion among nurses and supporting staff. In contrast, findings showed that nurses and supporting staff could be potential target groups for further trainings, as their scores were enhanced the most compared with other health professionals.” I do not quite agree with this assessment of the findings. Firstly, without knowing how the participants were selected for the training (did they self-select, in which case, they might be already more open to abortion), and without seeing the levels pre-training, it is impossible to state whether the findings presented here support or contradict these previous findings.
Author Response
- Some important information is missing, especially those related to the data collection procedure. I also recommend to better describe the training program.
Data was obtained from the WHRRF. Researchers in this study did not directly involved in conducting the training hence there is no biases. Additional details on the training have been added. Please see Line 99-103.
- It is also necessary a better description of the ethical considerations. Did the participants know that their information would be anonymized? Did they accept to participate in the study?
Participants had applied to the training. However, this training was done as part of the routine capacity building programmes of the WHRRF, where the training modules, contents and processes were endorsed by the Royal College of Obstetrics and Gynecology of Thailand. Besides, this work is part of the monitoring progress on access to sexual and reproductive health services, conducted by International Health Policy program (IHPP), a policy research institute under the Ministry of Public Health. In such circumstances, ethics application was not required. Please see Line 310-314.
- The results show differences between doctors and nurses. Has it also been studied gender differences?
In this study, 15% of participants were male and 84.6% were female. As shown in Table 3, participant gender had no significant association with the change in attitude or responses.
- It also would be interesting to include in the discussion the area in which the study was developed as well as the differences that other studies documented between rural and urban areas.
The analysis on attitude difference between urban and rural areas was not conducted, which poses a limitation to this study and warrants further investigation. The participants profiles did not capture whether they are providing services in urban or rural settings. This has been added to the discussion section. Please see Line 291-296.
Round 2
Reviewer 1 Report
More information on the data collection procedure and on the training of the professionals is necessary.
Author Response
We have added additional information on the training. Please see line 104-111. We have also added information on the data collection. Please see line 117-121. We would like to note that the data from the survey was obtained by the authors from the Women’s Health and Reproductive Rights Foundation of Thailand. As the survey and training was not administered by the authors, we have added the information requested to the Introduction and study design section.